# Association of Folate and Vitamins Involved in the 1-Carbon Cycle with Polymorphisms in the Methylenetetrahydrofolate Reductase Gene (MTHFR) and Global DNA Methylation in Patients with Colorectal Cancer

**DOI:** 10.3390/nu11061368

**Published:** 2019-06-18

**Authors:** Ariana Ferrari, Giovana Tardin Torrezan, Dirce Maria Carraro, Samuel Aguiar Junior

**Affiliations:** 1Department of Nutrition, Unicesumar, Av. Guedner, 1610, Jardim Aclimação, Maringá, Paraná CEP 87050-900, Brazil; 2Laboratory of Genomics and Molecular Biology, International Research Center, A.C. Camargo Cancer Center, Rua Professor Antônio Prudente, 211, Liberdade, São Paulo, São Paulo CEP 01509-010, Brazil; giovana.torrezan@accamargo.org.br (G.T.T.); dirce.carraro@accamargo.org.br (D.M.C.); 3Department of Pelvic Surgery, A.C. Camargo Cancer Center, Rua Professor Antônio Prudente, 211, Liberdade, São Paulo, São Paulo CEP 01509-010, Brazil; samuel.aguiar.jr@gmail.com

**Keywords:** folic acid, 1-carbon cycle, DNA methylation, colorectal carcinogenesis

## Abstract

Folate, vitamin B2, vitamin B6, vitamin B12, choline, and betaine are nutrients involved in the 1-carbon cycle that can alter the levels of DNA methylation and influence genesis and/or tumor progression. Thus, the objective of this study was to evaluate the association of folate and vitamins involved in the 1-carbon cycle and MTHFR polymorphisms in global DNA methylation in patients with colorectal cancer gene. The study included 189 patients with colorectal adenocarcinoma answering a clinical evaluation questionnaire and the Food Frequency Questionnaire (FFQ) validated for patients with colon and rectal cancer. Blood samples were collected for evaluation of MTHFR gene polymorphisms in global DNA methylation in blood and in tumor. The values for serum folate were positively correlated with the equivalent total dietary folate (total DFE) (rho = 0.51, *p* = 0.03) and global DNA methylation (rho = 0.20, *p* = 0.03). Individuals aged over 61 years (*p* = 0.01) in clinicopathological staging III and IV (*p* = 0.01) and with + heterozygous mutated homozygous genotypes for the MTHFR A1298C gene had higher levels of global DNA methylation (*p* = 0.04). The association between dietary intake of folate, serum folate, and tumor stage were predictive of global DNA methylation in patients’ blood. The levels of serum folate, the dietary folate and the status of DNA methylation can influence clinicopathological staging.

## 1. Introduction

Epigenetics is defined as inherited changes in gene expression, but without altering the base sequence of deoxyribonucleic acid (DNA), and may be reversible, unlike genetic alterations. The most studied epigenetic mechanisms are DNA methylation, histone modifications, and the action of non-coding ribonucleic acids (RNAs) [1]. Methylation consists of the covalent addition of a methyl carbon group at the 5-position of the pyrimidine ring of the cytosine nucleotide of DNA, which usually precedes a guanine (CG dinucleotides) [2]. DNA methyltransferase enzymes (DNMTs) are essential for this reaction and results in the formation of 5-methylcytosine, a methylated form of cytosine [3]. This epigenetic event is essential for gene regulation. However, hypomethylation or hypermethylation abnormalities are important in the genesis of tumors and are frequently found in the presence of neoplastic cells [4,5].

Folate and other B-complex vitamins have been identified as having a key role in the 1-carbon cycle, as they are essential nutrients for DNA synthesis and S-adenosylmethionine (SAM) [5]. The main enzymatic reaction of carbon transfer in humans occurs during the reversible conversion of serine to glycine. In this case, the hydroxymethyl group of the serine molecule is accepted by the FH4 molecule, resulting in glycine and 5,10-methylenetetrahydrofolate (5,10-MTHF). This molecule undergoes a reducing action of the enzyme 5,10 methylenetetrahydrofolate reductase (MTHFR) being converted to 5-MTHF, a folate form more present in the plasma and substrate responsible for the conversion of homocysteine to methionine. Methionine is subsequently metabolized to SAM, which will be responsible for DNA methylation [6]. In addition, the chemical structure of 5,10-MTHF, together with vitamin B6, is used for the methylation of the deoxythymidine component (dTMP) of the DNA molecule through deoxyuridine monophosphate (dUMP). The dTMP molecule, in turn, is essential for DNA synthesis [7,8].

DNA methylation is essential for the regulation of gene expression in the body, and an aberrant pattern of this methylation can lead to genomic instability and alter gene transcription, thus contributing to the development of various neoplasms [9]. According to SILVA et al. (2013), during colorectal carcinogenesis there is an association between global hypomethylation of DNA with hypermethylation in specific gene promoter regions that act on cell cycle regulation, DNA repair, apoptosis, angiogenesis, and cell adhesion [10,11,12].

Therefore, deficiencies of these vitamins may contribute to changes in SAM levels and DNA methylation. However, recent studies have shown that these vitamins act as an important substrate for tumor progression when an established pre-neoplastic lesion is already present. Thus, this work is justified by the need to know about the nutritional consumption of these nutrients in patients with pre-treatment colorectal adenocarcinoma, as well as to verify whether these nutrients’ influence the global DNA methylation, in the presence or not of important polymorphisms of the key gene for the metabolism of folate.

The aim of the present study was to assess the association between serum folate levels and polymorphisms in the MTHFR gene; and to identify and global DNA methylation in colorectal cancer patients.

## 2. Materials and Methods

This is a cross-sectional observational study with prospective data collection. The population of the present study was recruited consecutively from the admission of new cases of patients diagnosed with colon or rectal adenocarcinoma, referred to the Colorectal Tumor Service of A.C. Camargo Cancer Center (São Paulo, Brazil), from May 2011 to May 2012. Patients at any stage of the disease with indication of surgical intervention on the primary site were considered eligible.

Patients previously submitted to any kind of treatment at another institution; patients admitted with recurrent disease; or patients with history of another malignancy within the last six months were excluded. Patients whom were not able to understand the questionnaires or the informed consent form were also excluded. All patients included were staged and treated at A.C. Camargo Cancer Center. 

### 2.1. Ethical Aspects

This study was approved by the Research Ethics Committee of Fundação Antonio Prudente under number 1861/14.

### 2.2. Clinical Evaluation

The clinical evaluation record covered sociodemographic issues, personal and family history of cancer, associated pathologies, lifestyle habits, and intake of folic acid supplements. The clinicopathological staging was performed by the patient’s physician, according to the Cancer Staging Manual published by the American Joint Committee on Cancer (AJCC) [13]. A total of 195 patients answered the clinical evaluation questionnaire.

### 2.3. Dietary Assessment

An assessment was performed of the habitual dietary intake of alcohol, folate, vitamin B2, vitamin B6, vitamin B12, choline, betaine, methionine, energy, carbohydrate, protein, and lipid using a specific food frequency questionnaire (FFQ) for colorectal cancer patients, validated by LAMEZA (2010) [14], in the study “Validation of food frequency questionnaire for patients treated for colorectal cancer”. The portions of this questionnaire were elaborated based on habitual consumption of residents of the city of São Paulo, Brazil. The instrument was developed based on the R24h of 1477 adults, resulting in a questionnaire with 67 food items. After validation for patients with colorectal tumors, the questionnaire emerged into 110 food items [15]. 

The FFQ was applied by the researcher, after the first medical appointment where the diagnosis was made. During the FFQ, the patient was asked to remember the frequency of consumption of each food item in the last year (1 to 10 times a day, week, month, or year) and the portion size (small, medium, or large).

For tabulation of data, the selected portion of each food item was multiplied by the frequency of consumption and divided by the number of days (1, 7, 30, or 365 days). After that, these values were converted into energy and nutrient values through the Nutrition Data System for Research (NDSR) [16] software that uses data from the United States Department of Agriculture [17]. In relation to dietary folate, the separate values of natural folate and synthetic folate were first considered. Dietary folate values were corrected considering the mandatory fortification of wheat and corn flours (150 μg of folic acid/100 g of flour) that was implemented in Brazil since 2004 [18]. In addition, differences in the amount of folic acid addition in fortified foods in Brazil, of 150 μg/100 g of flour, and of the United States of America, of 140 μg/100 g of flour were considered. After these corrections, folate diet equivalent (FDE) values were calculated, taking into account differences in bioavailability of naturally occurring folate in foods and fortified folic acid in foods (1 μg of synthetic folate = 1.7 μg of folate) [19]. 

In addition to the dietary FDE values, folic acid intake through supplementation (synthetic folate supplement) was considered. From this value, the FDE of the supplement was calculated, considering that every 1 μg of folic acid as supplement, on an empty stomach, offers 2.0 μg of FDE [20]. With the sum of dietary FDE and supplement FDE, the total FDE was calculated. In total, 20 patients used folic acid supplements. After that, the raw FFQ data were submitted to calibration in order to reduce the biases induced by the errors inherent to the measures of food consumption, according to the previous study by FERRARI et al. (2018) [15].

### 2.4. Serum Folate

For the assessment of serum folate, 10 mL of blood was collected from the patients in 4 h of fasting by venipuncture and in the preoperative period. For patient candidates eligible for neoadjuvant treatment, samples were taken from peripheral blood before radiochemotherapy. The serum was obtained by a centrifugation method and the competitive enzyme immunoassay technique was used to analyze the serum concentration of folic acid [21]. 161 patients underwent the serum folic acid test.

### 2.5. MTHFR Gene Polymorphism

DNA samples were obtained from peripheral blood and the DNA was extracted from 8 mL of whole blood (4 mL in each EDTA tube) using the QIAamp DNA Blood Mini QIAcube Kit (QIAamp Blood) (QIAGEN, Valencia, CA, USA) following the manufacturer’s instructions. For the evaluation of C677T and A polymorphisms1298C in the MTHFR gene, the collected samples were submitted to direct sequencing. Sequencing reactions were performed with 2.5 μL purified PCR product, 1 μL BigDye Terminator v3.1 (Applied Biosystems, Foster City, CA, USA), 1.5 μL buffer, and 0.25 μM each specific primer in a final volume of 10 μL. The resulting sequences were analyzed by the CLC Bio software and compared to the MTHFR gene sequence.

### 2.6. Global DNA Methylation

The overall methylation of the DNA was evaluated based on the assumption that DNA methylation is essential for gene regulation and that hypomethylation or hypermethylation abnormalities are important in the genesis of tumors and are frequently found in the presence of neoplastic cells [22].

Global methylation of tumor and blood genomic DNA was evaluated using an enzyme-linked immunosorbent assay (ELISA) for the detection of methylated cytosine antibodies present in a given sample. Genomic DNA was quantified using Qubit™ dsDNA HS Assay Kit (Life Technologies, Carlsbad, CA, USA). 200 ng of genomic DNA was used for the methylation quantification reaction using the Imprint^®^ Methylated DNA Quantification Kit (MDQ1-96RXN—Sigma, St. Louis, MO, USA) following the manufacturer’s instructions. Briefly, steps of DNA binding to the plate surface were performed, followed by multiple steps of washing and binding of the capture and detection antibodies. After the colorimetric reaction, the amount of methylated DNA present in the sample is measured by the absorbance (450 nm) of each sample in microplate spectrophotometry (SpectraMax M5 Multi-Mode Microplate Reader).

A standard curve was performed on each reaction plate using five dilutions of a methylated control DNA provided in the kit (100 ng, 50 ng, 25 ng, 12.5 ng, and 6.25 ng of control DNA). The patient samples were evaluated in duplicate and the mean absorbance value of each duplicate was used to calculate the amount of methylation relative to the methylated control DNA by a linear regression calculation (considering the five dilutions of the control as 100%, 50%, 25%, 12.5%, and 6.25% methylated).

### 2.7. Statistical Analysis

Statistical analysis was performed using the software Statistical Analysis System (SAS) version 9.3 [23]. The level of significance considered in the analyses were 5%. For the test of data normality, the Shapiro–Wilk test for data normality was applied.

Fisher’s exact test and the chi-square test were used to compare the C677T and A1298C polymorphisms according to sociodemographic and clinical variables. In order to evaluate the relationship between polymorphisms and folate plasma levels, ANOVA and *t* tests were used.

For the association of DNA methylation with the variables age, tumor location, clinicopathological staging, and supplementation, a Student’s *t*-test was used. The ANOVA test was used for the association with color, educational level, polymorphisms, DFE diet, DFE total, vitamin B2, vitamin B6, methionine, choline, betaine, and alcohol. Prior to applying each *t*-test, a variance test was performed. In addition, if a significant difference between treatments was found, a Tukey’s test was used to test any contrast between two means. The Pearson’s test was applied to evaluate the association between serum folate and global DNA methylation. In addition, a covariance analysis was performed with DFE diet, staging, and serum folate to investigate the influence of these variables on global DNA methylation.

## 3. Results

During this period, 195 patients met the inclusion criteria and were invited to participate. Among them, 189 filled out the FFQ: 161 had serum folate measured, 128 had MTHFR polymorphisms analyses, and 123 had blood samples adequate for DNA methylation analyses. Considering the 189 patients with fulfilled FFQ, the mean age was 61 years. Concerning gender, 93 were female (49.2%) and 96 were male (50.8%) (Table 1).

There was no significant difference between serum folate levels in relation to the different genotypes of the polymorphisms in the MTHFR gene (Table 2).

Table 3 shows the association between the sociodemographic variables and global DNA methylation in the tumor and blood of the patient. It was observed that patients aged 61 years and older had overall tumor DNA methylation levels greater when compared to individuals younger than 61 years (*p* = 0.01).

Regarding the clinical variables, patients with more advanced clinicopathological staging had significantly higher levels of global methylation in the blood compared to patients with stage I or II (*p* = 0.01) (Table 4). It was also verified whether the joint association between the polymorphisms (C677T and A1298C) in the MTHFR gene interfered with DNA methylation, but no significant association was found. However, in the evaluation of the A1298C polymorphism, when wild-type homozygous (AA) versus mutated homozygote + heterozygote (CC + AC) genotypes were compared, patients with wild-type homozygous genotypes had higher statistically significant levels of global DNA methylation in the tumor compared to mutated homozygote + heterozygote patients (*p* = 0.04).

There was no significant association between these variables and global DNA methylation (Table 5 and Table 6). However, when the correlation between serum folate and DNA methylation was evaluated, a weak but significant positive correlation was observed between plasma folate levels and global DNA methylation in the blood (rho = 0.20 and *p* = 0.03) (Table 7).

For the evaluation of DNA methylation predictors, a covariance analysis was performed, according to Table 7. It is noted that the association between dietary intake of folate, serum folate, and tumor staging were predictive of global DNA methylation in the blood of patients.

## 4. Discussion

As a vitamin involved in multiple biochemical processes, folate has been studied as an important modulator of carcinogenesis [24,25,26]. Folic acid is an essential nutrient for the 1-carbon cycle, which at one time participates in the synthesis of nucleotides and methylation reactions in the human organism [27]. Together with the nutrients involved in the 1-carbon cycle, the MTHFR enzyme plays an important role during the synthesis, repair and methylation of DNA, and may promote changes in circulating folate levels [28]. However, in our study, no association was found between polymorphisms and serum folate levels, which corroborates the work of Hanks et al. (2013) [29] but differ from Heavey et al. (2004) [30] and Mei et al. (2012) [31].

In the present study as to the age of patients older than 61 years, presented greater global methylation levels in the tumor than patients of lower age. This data, corroborated by the study of Wallace et al. (2010), who evaluated 1000 biopsies of normal colorectal mucosa in 389 patients, found an increase in methylation in the “CpG islands” according to the increase in the individuals’ age [32]. In contrast, Bollati et al. (2009) assessed DNA methylation in 1097 blood samples in 718 healthy subjects aged 55 to 92 years and found a progressive loss of DNA methylation in repetitive elements throughout the genome [33].

In our study, the polymorphism in the *MTHFR C677T* gene was not significantly associated with DNA methylation. Diversely, Castro et al. (2004), in their study with 96 healthy individuals, showed a lower methylation rate in individuals with TT genotype for the *MTHFR* C677T polymorphism when compared to subjects with CC genotype [34]. In addition, other researchers found that individuals with lower levels of serum folate and TT genotype had a lower methylation rate than those with low serum folate, but with CC genotype [35]. Some studies affirm that low folate levels associated with *MTHFR C677T* gene polymorphism (TT genotype) are associated with hypomethylation of DNA [36,37]. Narayanan et al. (2004) in their study found no association between the *MTHFR* C677T polymorphism and global DNA methylation, in keeping with the current study [38].

Regarding the *MTHFR* A1298C polymorphism, patients with mutated homozygous genotypes associated with heterozygotes (CC + AC) had lower levels of global DNA methylation in the tumor than patients with wild-type genotype (AA). The association was weak but significant. The CC and AC genotypes are associated with the decrease of the MTHFR enzyme to a lesser and higher degree, respectively, when compared to the wild-type homozygous genotype. In fact, some studies have shown that this polymorphism may decrease the activity of the MTHFR enzyme and lead to a state of DNA hypomethylation [39,40]. One fact that drew attention was the association between clinicopathological staging and the global DNA methylation in blood, and patients with more advanced staging had higher levels of global DNA methylation. However, no studies were found in the literature evaluating such an association in patients undergoing pre-treatment for colorectal cancer. For this reason, only a few hypotheses will be presented for this finding.

Aberrant DNA methylation has been studied as an important epigenetic alteration in colon and rectal neoplasia. It is responsible for a series of molecular changes in DNA and in chromatin structure, which influences genes that promote tumor initiation and progression [41]. Often in normal cells, regions with CpGs distributed by the genome are found methylated and the “CpGs islands” hypomethylated. The unmethylated region shows an open chromatin structure and active transcription for gene regulation. An example is the promoter region of the tumor suppressor gene, which is hypomethylated in normal cells [42].

However, during tumor development, studies point to two important epigenetic alterations. First, hypermethylation of promoter regions rich in CG dinucleotides lead to a silencing of tumor suppressor genes [43]. The silencing mechanism is possibly due to a lack of binding of transcription factors and changes in the chromatin structure, since the methylated DNA recruits complexes of proteins that promote the deacetylation of the histones, leading to the compaction of the chromatin. Thus, DNA methylation represses transcription by inhibiting binding of transcription factors and by recruitment of proteins that bind to methylated CpG [41]. The silencing of tumor suppressor genes can promote the initiation of carcinogenesis and neoplastic progression by inhibition of apoptosis and autophagy pathways in cancer cells [44]. In addition, it is known that when the cytosine molecule is methylated and converted to 5-methylcytosine, it is unstable and may undergo spontaneous deamination, being converted into thymine. This type of point mutation (C-G for T-A) may influence the function of oncogenes or tumor suppressor genes [45]. Secondly, global hypomethylation of DNA is observed, interfering with genomic stability, reactivation of transposable elements, and loss of gene imprinting patterns, thus contributing to tumor development [41,46,47]. In addition to these mechanisms, some studies suggest that alterations in epigenetic mechanisms may influence the epithelial-to-mesenchymal transition, through the silencing of integrins, and consequently tumor progression. It is also known that epigenetic mechanisms may interfere with miRNA, which are involved with tumor development and progression [48].

In the present study, patients with more advanced staging had higher levels of global DNA methylation when compared to patients in earlier stages. It is already well established that the global hypomethylation of DNA is a frequent and early characteristic of carcinogenesis, listed with the prognosis of several types of tumors. However, no studies were found to evaluate methylation levels in the different stages of colorectal cancer. Wei et al. (2002), evaluated the global DNA methylation of 19 patients in stages III and IV with ovarian carcinoma. As a result, patients in the group with global DNA hypermethylation were associated with worse prognosis and recurrence after chemotherapy when compared to patients with lower levels of methylation (*p* < 0.001). The study suggests epigenetic markers to predict the outcome of treatment in ovarian cancer patients [49], while another case-control study has not found significant difference between global DNA methylation and staging in different groups of patients [50].

Thus, at the onset of tumorigenesis, the presence of global DNA hypomethylation that can lead to genomic instability is well established. However, as found in the present study in more advanced clinicopathological stages, one of the possible hypotheses would be the global hypermethylation of the DNA being the cause of the more advanced staging or the consequence of the same. Another important point concerns the methodology used for the evaluation of DNA methylation. Many studies analyze global DNA methylation in repetitive elements such as LINE-1, a methodology that is different from the present work, which may have influenced the different results found.

Clinicopathological staging and serum folate levels were predictive variables of this epigenetic event. However, no other studies were found with patients undergoing pre-treatment by colorectal cancer that evaluated these variables as being predictors of global DNA methylation.

Despite the protective action of folic acid against CCR [51], some studies call attention because of its double role, assuming that the folate intake, through an already established pre-neoplastic lesion, can accelerate the growth of tumor cells [50].

Some theories try to explain the double role of folic acid during colorectal carcinogenesis. Firstly, it is important to emphasize that adequate folate intake is associated with numerous health benefits, which encompass reduction of neural tube defects, reduced risk of cardiovascular diseases and cancer. However, in individuals without pre- or neoplastic lesions, low folate intake and decreased serum folate levels have been associated with increased risk of colorectal tumors [51]. The mechanism by which folate deficiency can lead to colorectal cancer is related to synthesis of purines and thymidylate. Thus, adequate levels are essential for the synthesis, stability, integrity, and adequate repair of DNA. In addition, folate deficiency is also associated with chromosomal instability, changes in transcriptional regulation, poor incorporation of the uracil into DNA, and hypomethylation of this molecule [52].

However, in cells with intense cell division, folic acid appears to act differently. First, based on the assumption that pre-neoplastic lesions or neoplastic cells have high replication and proliferation rates, high levels of folate could favor the synthesis and replication of DNA through nucleotide synthesis. Thus, in neoplastic cells, folic acid deficiency could lead to a change in their metabolism and DNA synthesis, thus resulting in inhibition of tumor growth and even tumor regression [53].

Kim (2004), analyzing animal studies and clinical observations, focused on the double role of folate in carcinogenesis, showing that this nutrient in doses above the recommended level may increase disease progression when pre-lesion is already present. In fact, some authors point out that folate intake above the daily nutritional recommendation in addition to not reducing the risk for colorectal cancer but also increases the risk of an individual developing this type of tumor and even the recurrence of adenomas [54]. Concern increases when reminded of the mandatory fortification of folic acid in food, which is the most important action in the field of nutrition and public health [55].

However, Wolpin et al. (2008) evaluated the serum folate of 301 patients with colorectal tumors, in order to verify if this nutrient influences the progression of these neoplasms. As in the present study, patients who consumed a greater amount of dietary folate and who drank dietary supplements regularly had higher plasma levels of folic acid [56].

The aberrant methylation of “CpG islands” can be considered an important tumor cell proliferation factor, leading to a silencing of tumor suppressor genes and molecules involved in cell differentiation. Hypermethylation, often observed in sporadic cases of cancer, is an event which can target several genes simultaneously, some of which are responsible for tumor genesis and/or progression [57]. More than half of the colorectal cancer present hypermethylation in at least one promoter region of tumor suppressor genes (DAPK—gene associated to apoptosis linked to protein kinase; p16INK4a and p14ARF—genes involved in the cell cycle), repair genes (MGMT-O6-methylguanine-DNA methyltransferase; hMLH1-human homolog of bacterial MutL) and genes involved in progression and metastases (TIMP3, CXCL12, ID4, and IRF8) [56]. Some studies, for example, show that hypermethylation of “CpG islands” in the promoter region of the gene may lead to loss of MGMT [58]. In addition, DNA repair mechanisms can be affected according to DNA methylation status and nucleotide availability. Thus, increased rates of folic acid may lead to hypermethylation of tumor suppressor genes and other antineoplastic genes, inactivating them, and promoting tumor progression [59].

Supplementation with high doses of folic acid is able to increase DNA methylation. A randomized study, in which 20 patients with adenomas received or not folic acid supplementation for 1 year after polypectomy found that folate supplementation increased both serum folate levels and the degree of DNA methylation, with the latter observed at 6 months for patients with supplementation and 1 year for the placebo group [60]. Other similar studies concluded that folic acid supplementation increases serum folate levels and DNA methylation [61].

Thus, epigenetic events are increasingly being studied as they do not reach the altered gene information and are therefore potentially reversible. DNA methylation is considered to be the most important epigenetic phenomenon because it is essential for normal gene expression, DNA maintenance and integrity, chromosomal modifications, as well as for the appearance of mutations. This event occurs when the DNA methyl transferase enzyme transfers a methyl grouping of the SAM molecule to cytosine residues of the “CpG islands”.

In the last decade, several animal and epidemiological studies have reported beneficial effects of folate during carcinogenesis in different organs. However, recent studies have shown concern about the double role that folic acid can play, depending on the dose and the moment that the nutrient is ingested [62]. In our study we found that folic acid intake interferes with serum folate levels. In addition, folate intake, serum folate levels, global DNA methylation and age were predictors of clinicopathological staging. Although the present study was not designed to evaluate DNA methylation as a biomarker of metastasis, the data found here may stimulate further investigations to better understand this association and DNA methylation as a potential biomarker of tumor progression.

## Figures and Tables

**Table 1 nutrients-11-01368-t001:** Characteristics of 189 patients with fulfilled Food Frequency Questionnaire (FFQ).

Clinical Characteristics	*n* (%)
**Gender**	
Female	93 (49.2)
Male	96 (50.8)
**Ethnicity (self-reported)**	
Asian	35 (18.5)
White	135 (71.4)
Mulatto/black	19 (10)
**Tumor location**	
Colon	125 (66.1)
Rectum	64 (33.9)
**Stage**	
NA	3 (1.6)
I–II	97 (51.3)
III–IV	89 (47.1)
**Total**	**189**

**Table 2 nutrients-11-01368-t002:** Association between the levels of serum folate and 5,10 methylenetetrahydrofolate reductase (MTHFR) gene polymorphisms.

MTHFR Gene Polymorphisms	Serum Folate Levels		
	*n*	Mean (DP)	*p*
C677T polymorphisms			
CC	45	13.31 (±6.05)	0.27
CT	43	11.82 (±5.92)	
TT	20	10.97 (±5.76)	
A1298C polymorphisms			
AA	70	12.17 (±5.81)	0.96
AC	31	12.45 (±6.39)	
CC	7	12.59 (±6.44)	

**Table 3 nutrients-11-01368-t003:** Association between the levels of global DNA methylation and the sociodemographic variables of patients undergoing pre-treatment for colorectal adenocarcinoma.

Sociodemographic Variables	Global DNA Methylation in Tumor (%)			Global DNA Methylation in Blood (%)		
	*n*	Mean (SD)	*p*	*n*	Mean (SD)	*p*
Age	111			123		
<61 years	49	46.05 (±17.00)	0.01 *	52	28.20 (±23.63)	0.59
>=61 years	62	56.47 (±27.95)		71	26.23 (±14.86)	
Color	111			123		
Yellow	21	61.72 (±32.83)	0.10	24	27.34 (±11.16)	0.35
White	79	50.02 (±21.54)		86	28.06 (±21.72)	
Mulatto/Black	11	46.35 (±20,021)		13	19.90 (±5.18)	
Education level	111			123		
Complete primary education	39	51.84 (±23.23)	0.81	43	24.38 (±9.71)	0.47
Incomplete secondary education	33	49.88 (±19.29)		34	27.39 (±18.75)	
Superior/Graduate complete	46	29.33 (±24.92)		39	53.59 (±28.94)	

SD = standard deviation; * *p* < 0.05.

**Table 4 nutrients-11-01368-t004:** Association between the levels of global DNA methylation and the clinical variables of patients undergoing pre-treatment for colorectal adenocarcinoma.

Clinical Variables	Global DNA Methylation in Tumor (%)			Global DNA Methylation in Blood (%)		
	*n*	Mean (SD)	*p*	*n*	Mean (SD)	*p*
Tumor location	111			123		
Colon	83	53.12 (±23.55)	0.35	92	25.60 (±13.23)	0.30
Rectum	28	48.16 (±26.18)		31	31.39 (±30.20)	
Clinicopathological staging	111			123		
I and II	58	50.15 (±23.49)	0.43	68	23.29 (±8.17)	0.01 *
III and IV	53	53.76 (±25.07)		55	31.73 (±26.32)	
Polymorphism C677T	102			123		
CC	43	53.77 (±25.21)	0.84	50	27.21 (±16.15)	0.91
CT	42	51.53 (±24.35)		49	27.61 (±24.57)	
TT	17	55.28 (±25.60)		24	25.62 (±9.84)	
Polymorphism A1298C	102			123		
AA	62	56.72 (±27.42)	0.11	78	27.15 (±20.27)	0.74
AC	33	49.19 (±19.69)		37	27.92 (±18.26)	
CC	7	39.44 (±11.41)		8	22.19 (±5.34)	
Polymorphism C677T	102			123		
CC ^a^	43	53.77 (25.21)	0.81	50	27.21 (±16.15)	0.94
TT + CT ^b^	59	52.61 (24.55)		73	26.96 (±20.84)	
Polymorphism A1298C	102			123		
AA ^a^	62	56.72 (27.42)	0.04 *	78	27.15 (±20.27)	0.94
CC + AC ^b^	40	47.48 (18.77)		45	26.90 (±16.80)	

SD = standard deviation; ^a^ wild-type homozygous; ^b^ mutated homozygous + heterozygous; * *p* < 0.05.

**Table 5 nutrients-11-01368-t005:** Association between levels of global DNA methylation with dietary intake of folate, alcohol, and methionine of patients undergoing pre-treatment for colorectal adenocarcinoma.

Dietary Variables (Folate, Alcohol, and Methionine)	Global DNA Methylation in Tumor (%)			Global DNA Methylation in Blood (%)		
	*n*	Mean (SD)	*p*	*n*	Mean (SD)	*p*
Supplementation	111			123		
No	99	52.02 (±24.60)	0.85	111	27.13 (±19.89)	0.81
Yes	12	50.68 (±21.65)		12	26.47 (±6.97)	
DFE diet ^1^	111			123		
1st tertile	35	49.82 (±22.36)	0.26	44	32.29 (±28.96)	0.06
2nd tertile	40	56.83 (±26.41)		40	24.86 (±9.08)	
3rd tertile	36	48.36 (±23.11)		39	23.42 (±8.68)	
DFE total ^2^	12			12		
1st tertile	1	34.22	0.71	1	25.44	0.34
2nd tertile	5	49.44 (±13.25)		5	30.03 (±7.54)	
3rd tertile	6	54.46 (±28.61)		6	23.67 (±6.24)	
Alcohol	111			123		
1st tertile	36	51.81 (±22.65)	0.96	37	24.67 (±8.42)	0.64
2nd tertile	41	52.54 (±25.46)		46	27.65 (±25.55)	
3rd tertile	34	51.13 (±24.97)		40	28.59 (±17.90)	
Methionine	111			123		
1st tertile	34	54.89 (±25.90)	0.65	38	31.59 (±30.76)	0.15
2nd tertile	38	51.43 (±26.22)		45	26.49 (±9.80)	
3rd tertile	39	49.67 (±20.79)		40	23.40 (±9.48)	

SD = standard deviation; DFE = dietary folate equivalent; ^1^ DFE diet = natural folate + 1.7 × (synthetic folate in diet); ^2^ DFE total = DFE diet + DFE supplement. Note: The first tertile represents the lower level of nutrient intake and the last tertile (3°) the highest level of nutrient intake.

**Table 6 nutrients-11-01368-t006:** Association between the levels of global DNA methylation with the dietary intake of the nutrients involved in the 1-carbon cycle of patients undergoing pre-treatment for colorectal adenocarcinoma.

Dietary Variables	Global DNA Methylation in Tumor (%)			Global DNA Methylation in Blood (%)		
	*n*	Mean (SD)	*p*	*n*	Mean (SD)	*p*
Vitamin B2	111			123		
1st tertile	36	52.42 (±24.25)	0.43	41	28.70 (±17.73)	0.52
2nd tertile	33	55.66 (±24.32)		38	28.31 (±27.14)	
3rd tertile	42	48.42 (±24.19)		44	24.46 (±9.35)	
Vitamin B6	111			123		
1st tertile	30	53.60 (±29.67)	0.84	40	31.03 (±30.04)	0.13
2nd tertile	41	50.26 (±19.86)		44	27.51 (±10.60)	
3rd tertile	40	52.23 (±24.27)		39	22.48 (±8.23)	
Vitamin B12	111			123		
1st tertile	36	53.49 (±26.52)	0.77	41	27.34 (±17.52)	0.50
2nd tertile	37	52.58 (±24.21)		42	29.30 (±26.01)	
3rd tertile	38	49.65 (±22.35)		40	24.42 (±9.55)	
Choline	111			123		
1st tertile	35	52.02 (±24.86)	0.68	40	32.70 (±29.83)	0.07
2nd tertile	39	54.19 (±25.69)		44	23.80 (±8.81)	
3rd tertile	37	49.29 (±22.32)		39	24.96 (±10.39)	
Betaine	111			123		
1st tertile	37	49.99 (±18.42)	0.83	46	29.07 (±24.89)	0.28
2nd tertile	36	53.24 (±27.85)		40	28.56 (±18.11)	
3rd tertile	38	52.41 (±25.95)		37	22.94 (±8.34)	

SD = standard deviation.

**Table 7 nutrients-11-01368-t007:** Covariance model and significance of predictors of global DNA methylation of patients undergoing pre-treatment for colorectal adenocarcinoma.

Global DNA Methylation	*p*
in tumor	
Model (DFE diet + serum folate + tumor staging)	0.37
DFE diet ^1^	0.41
Tumor staging Serum folate	0.39 0.24
in blood	
Model (DFE diet + serum folate + tumor staging)	0.01 *
DFE diet	0.15
Tumor staging Serum folate	0.03 * 0.02 *

DFE = dietary folate equivalent; ^1^ DFE diet = natural folate + 1.7 × (synthetic folate in diet); * *p* < 0.05.

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
