# Peer review of "Association of Folate and Vitamins Involved in the 1-Carbon Cycle with Polymorphisms in the Methylenetetrahydrofolate Reductase Gene (MTHFR) and Global DNA Methylation in Patients with Colorectal Cancer"

_nutrients, 2019, doi:10.3390/nu11061368_

Round 1
Reviewer 1 Report
The work is well presented and shows that eating habits together with other factors can represent risk factors for epigenetic events and moreover a genetic predisposition represented by polymorphism together with exogenous factors can predispose the subject to develop a tumor and have a worse prognosis . Why was the analysis conducted on only 123 patients? What happened to the other 72 patients? It would have been interesting to know the patients' eating habits, such as those who were vegetarians, those who followed a more protein diet, how many a Mediterranean-type diet. Why was this information not reported? Furthermore, the multivariate analysis is missing.
Author Response
Response to Reviewer 1 Comments
Point 1: Why was the analysis conducted on only 123 patients? What happened to the other 72 patients?
Response 1:
We have missed one or more measurements among these 72 patients.
During this period, 195 patients met the inclusion criteria and were invited to participate. Among them, 189 filled the food frequency questionnaire (FFQ); 161 had serum folate measured; 128 had MTHFR polymorphisms analyses; and 123 had blood samples adequate for DNA methylation analyses.
Point 2: It would have been interesting to know the patients' eating habits, such as those who were vegetarians, those who followed a more protein diet, how many a Mediterranean-type diet. Why was this information not reported?
Response 2:
It would be very interesting. Unfortunately, we have not collected these data prospectively, as our structured questionnaire did not include this variables. A retrospective collection would be very questionable. So, we are sorry about not including these data on the analysis.
Point 3: The multivariate analysis is missing.
Response 2: As we had different types of data, our option was for performing models of covariance, instead of another models of multivariate regression.
Note: The changes made in the paper are in red color.
Reviewer 2 Report
I enjoyed reading your well-edited manuscript however, further detail in needed in your introduction and methods.
Introduction
-Could you provide a diagram of the one-carbon cycle with the relevant nutrients? I think this would provide greater clarify for the readers.
-I would suggest spending some time strengthening the introduction section. Can you build a stronger case as to why your study is important? Can you review what others studies have found? What gaps in the literature is your study addressing?
Methods
-Colorectal Tumor Center of A.C. Camargo Cancer Center- Can you indicate the city and country of this center?
-The clinical evaluation questionnaire- Who developed this questionnaire? Did the patients complete the questionnaire themselves or was it administered by researchers?
-Can you indicate who diagnosed colon and rectal adenocarcinoma in the patients? Were all stages of the cancer included in the study? Were they receiving treatment?
-Can you provide an inclusion and exclusion criteria for the study?
-For the food frequency questionnaire- can you describe the population in the validation study (i.e. adults, pre-treatment for cancer). Was the FFQ measuring food intake from the previous six months? Who adminstered the FFQ? Did the FFQ collect information about vitamin and mineral supplement use? At what stage of the patient's cancer did they complete the FFQ?
-Can you please provide a section in your methods regarding dietary analysis? How was nutrient intake quantified from the FFQs? What software and databases were used?
-Why were serum measurements taken only for folate?
-QIAamp DNA Blood Mini QIAcube Kit- Please indicate the manufacturer and city and country of manufacturer in brackets.
-Were the blood samples held in storage for a period of time?
-Why was study only limited to global DNA methylation? Please indicate in your methods.
Results
-Can you indicate the mean age overall for your study sample? Can you indicate the percentage of females and males in your study?
-Can you indicate the direction of the tertile? Is the first tertile the highest level of nutrient intake?
-Did gender have any influence on the results?
-Further description of your results is needed.
Discussion
-Be careful with your use of the terms folate and folic acid
-Was expecting to see some discussion about other nutrients analysed in the study apart from folate
Author Response
Response to Reviewer 2 Comments
Point 1: Introduction - I would suggest spending some time strengthening the introduction section. Can you build a stronger case as to why your study is important? Can you review what others studies have found? What gaps in the literature is your study addressing?
Response 1: Thanks for the suggestion. In the introduction some information was included according to the suggestion.
Point 2: Methods - Colorectal Tumor Center of A.C. Camargo Cancer Center- Can you indicate the city and country of this center?
Response 2: This information has been added in the text.
Point 3: Methods - The clinical evaluation questionnaire- Who developed this questionnaire? Did the patients complete the questionnaire themselves or was it administered by researchers?
Response 3: This is not exactly a structured questionnaire. It was a registry developed and filled by the researches with clinical, pathological, sociodemographic, and others variables of interest. We agree that the term questionnaire is innapropriate, and we changed for clinical evaluation record.
Point 4: Methods - Can you indicate who diagnosed colon and rectal adenocarcinoma in the patients? Were all stages of the cancer included in the study? Were they receiving treatment? Can you provide an inclusion and exclusion criteria for the study?
Response 4: As AC Camargo is a reference center, patients use to be referred already with diagnosis of cancer. All patients were staged and treated at AC Camargo Cancer Center. We re-wrote a session in Methods describing inclusion and exclusion criteria.
Point 5: Methods - For the food frequency questionnaire- can you describe the population in the validation study (i.e. adults, pre-treatment for cancer). Was the FFQ measuring food intake from the previous six months? Who adminstered the FFQ? Did the FFQ collect information about vitamin and mineral supplement use? At what stage of the patient's cancer did they complete the FFQ?
Response 5: This information has been added in the text.
Point 6: Methods - Can you please provide a section in your methods regarding dietary analysis? How was nutrient intake quantified from the FFQs? What software and databases were used?
Response 6: This information has been added in the text.
Point 7: Methods - Why were serum measurements taken only for folate?
Response 7: Only serum folate was evaluated starting from the assumption that this vitamin is the most important involved in the carbon 1 cycle.
Point 8: Methods - QIAamp DNA Blood Mini QIAcube Kit- Please indicate the manufacturer and city and country of manufacturer in brackets.
Response 8: This information has been added in the text.
Point 9: Methods - Were the blood samples held in storage for a period of time?
Response 9: Yes. Blood, as well as tumor samples were frozen and stored at our institutional Tumor Bank for posterior analyses. We can not precisely describe how much time the samples remained stored. As the collection of data spent one year and the lab analyses were performed just after, we estimate that samples could be stored at a maximum of 18 months.
Point 10: Methods - Why was study only limited to global DNA methylation? Please indicate in your methods.
Response 10: This information has been added in the text.
Point 11: Results - Can you indicate the mean age overall for your study sample? Can you indicate the percentage of females and males in your study?
Response 11: A table with sociodemographic and clinical characteristics was added.
Point 12: Results - Can you indicate the direction of the tertile? Is the first tertile the highest level of nutrient intake?
Response 12: The first tertile represent the lower level of nutrient intake and the last tertile (3°) the highest level of nutrient intake..
Point 13: Results - Did gender have any influence on the results?
Response 13: The gender didn't influence on the results.
Point 14: Results - Further description of your results is needed.
Response 14: This information has been added in the text.
Point 15: Discussion - Be careful with your use of the terms folate and folic acid
Response 15: Terms have been revised.
Point 16: Discussion - Was expecting to see some discussion about other nutrients analysed in the study apart from folate
Response 16: As the main objective of our work was to evaluate the relationship of folate with the genetic events, it was focused on this nutrient in the discussion so that it does not exceed the maximum limit of pages.
Note: The changes made in the paper are in red color.

Round 2
Reviewer 1 Report
The paper has been correctly modified according to the indications of the reviewers.